



# Phytoplankton blooms affect microscale gradients of oxygen and temperature across the sea surface microlayer

Carsten Rauch[1], Lisa Deyle[1], Leonie Jaeger[1], Edgar Fernando Cortés-Espinoza[1], Mariana Ribas-Ribas[1], Josefine Karnatz[2], Anja Engel[2], Oliver Wurl[1]

[1] Center for Marine Sensors (ZfMarS), Institute for Chemistry and Biology of the Marine Environment (ICBM), School of Mathematics and Science, Carl von Ossietzky Universität Oldenburg, Ammerländer Heerstraße 114-118, 26129 Oldenburg, Germany
[2] GEOMAR Helmholtz Centre for Ocean Research Kiel, Wischhofstraße 1-3, 24148 Kiel, Germany
*Correspondence to*: Carsten Rauch (carsten.rauch1@uni-oldenburg.de)

**Abstract.** The sea surface microlayer (SML) is the thin layer on top of the ocean that is in direct contact with the atmosphere and is crucial for air–sea interactions. Its properties are influenced in particular by surface-active substances (surfactants), mainly produced by phytoplankton and bacteria. Thus, phytoplankton blooms and their decay can have a considerable influence on the SML. A mesocosm study was conducted to assess the impact of a phytoplankton bloom on the SML using a multidisciplinary approach, which enabled in situ measurements under controlled yet natural conditions. A phytoplankton bloom was induced within a mesocosm facility filled with seawater, resulting in three phases of the study: the pre-bloom, bloom, and post-bloom phases. During all phases, microsensors measured in situ microprofiles of oxygen and temperature with a 125 µm vertical resolution through the air, SML, and underlying water. Oxygen and temperature gradients were determined from the profiles, as well as the thicknesses of the oxygen diffusion boundary layer (DBL) and thermal boundary layer (TBL). The night-time oxygen gradients ($\emptyset_{\Delta O2,\ pre-bloom}$ = −2.16 ± 5.53 µmol L$^{-1}$, $\emptyset_{\Delta O2,\ bloom}$ = +24.90 ± 14.51 µmol L$^{-1}$, $\emptyset_{\Delta O2,\ post-bloom}$ = +2.07 ± 4.82 µmol L$^{-1}$) correlated highly with the chlorophyll *a* concentration (r = 0.755, p < 0.001), while the DBL thickness ($\emptyset_{DBL,\ overall}$ = 937 ± 369 µm) showed a moderate correlation to the SML surfactant concentration (r = 0.490, p = 0.014). Both indicate the phytoplankton bloom's influence on oxygen gradients across the SML. Night-time temperature gradients ($\emptyset_{\Delta T,\ overall}$ = −0.133 ± 0.079 °C) and the TBL thickness ($\emptyset_{TBL,\ overall}$ = 1300 ± 392 µm) were not correlated to the chlorophyll *a* or surfactant concentration. The mesocosm study and the microprofiling approach provide in situ data on the air–sea exchange processes in the SML, reflecting the distinct interplay of the SML and phytoplankton blooms in the exchange of oxygen and heat. This has implications for future studies on air–sea gas and heat exchange between the ocean and the atmosphere.

## 1 Introduction

The sea surface microlayer (SML) is of global importance as the layer that covers all oceans. Less than 1 mm thick, it forms the direct boundary between the atmosphere and the ocean (Cunliffe et al., 2013). This thin boundary layer is critical for the exchange of heat, momentum, gases, freshwater, and aerosols (Cunliffe et al., 2013; Wurl et al., 2016; Engel et al., 2017; Wong and Minnett, 2018; Cronin et al., 2019; Gassen et al., 2024; Laxague et al., 2024). A distinct SML covers large areas of the



ocean under typical wind conditions and can therefore significantly influence processes such as global air–sea gas or heat
exchange (Wurl et al., 2011; Wurl et al., 2016; Wurl et al., 2017). It accumulates biomass, in particular surface-active
substances (i.e., surfactants) (Wurl et al., 2016), and serves as a unique habitat for microorganisms (Stolle et al., 2010; Cunliffe
et al., 2013) and as a nursery ground for several species like crustaceans, molluscs, and fishes (Gallardo et al., 2021).

Slicks are an extreme form of the SML, covering approximately 30 % of coastal and 11 % of oceanic regions (Romano, 1996).
They have a thicker viscous sublayer that dampens capillary waves and creates patches during calm water (Saunders, 1967;
Katsaros, 1980). Physical and biological drivers, such as wind speed and primary production, play a crucial role in the
formation and dispersion of slicks. The SML, enriched with surfactants including carbohydrates, proteins and lipids, harbours
aggregates and gel-like particles exhibiting biofilm-like properties (Cunliffe et al., 2013; Wurl et al., 2016). Surfactants are
produced by phytoplankton and heterotrophic bacteria in the SML or the underlying water (ULW), and are transported to the
SML through rising air bubbles or buoyant particles (Žutić et al., 1981; Wurl et al., 2009; Kurata et al., 2016; Wurl et al.,
2016). Heterotrophic activities in the SML are crucial for the formation and retention of slicks (Stolle et al., 2010; Wurl et al.,
2016). Events such as phytoplankton blooms can increase surfactant concentrations, thereby leading to the formation of slicks
(Sieburth and Conover, 1965; Wurl et al., 2018; Barthelmeß and Engel, 2022). By dampening surface waves, slicks
significantly reduce air–sea heat and gas exchange fluxes (Katsaros, 1980; Laxague et al., 2024), for example, reducing global
air–sea $CO_2$ fluxes by 19 % (Mustaffa et al., 2020).

Although the SML has distinct properties that differ from those of the ULW (Hunter, 1997), its thickness is difficult to
determine because it is typically less than 1 mm and strongly influenced by environmental factors influencing near-surface
turbulence, such as wind speed and surfactant concentration (Wurl et al., 2011; Wurl et al., 2017). In field studies, the SML
thickness is often defined operationally by the thickness of the layer that a sampling device can skim off the water surface.
This approach does not necessarily represent the real thickness, and sampled thicknesses depend on the sampling device and
vary between 10 µm and 250 µm (Falkowska, 1999). For the well-established glass plate method, sampled thicknesses agree
well with the SML thickness of 50 µm, proposed through changes in chemical surface properties, but may vary from 40 µm
to 100 µm (Harvey and Burzell, 1972; Zhang et al., 1998; Falkowska, 1999; Engel and Galgani, 2016).

Due to these difficulties in directly investigating the SML, other surface sub-layers are often considered. These sublayers
include the diffusion boundary layer (DBL) of gases, such as oxygen or $CO_2$, and the thermal boundary layer (TBL) of
temperature. Methods for determining the DBL thickness often rely on indirect measurements of gradients of dissolved gases.
Carbon dioxide flux measurements were used to infer that the DBL thickness is approximately 50 µm (Robertson and Watson,
1992). The more recent application of microsensors has allowed for direct measurements of the DBL thickness of oxygen
between 350 µm and 1100 µm (Rahlff et al., 2019; Adenaya et al., 2021). For determining the TBL, the SML temperature is
typically measured using infrared radiometers and compared to subsurface temperatures. These measurements enable the
calculation of both the surface temperature gradient and the theoretical TBL thickness (Saunders, 1967). Surface temperature
gradients can range from +1.5 °C to –0.6 °C, but under common oceanic conditions, a cool skin layer with gradients typically
between –0.1 °C and –0.2 °C is observed due to a net heat loss at the ocean's surface (Ewing and McAlister, 1960; Robertson



and Watson, 1992; Donlon et al., 1999; Murray et al., 2000; Donlon et al., 2002; Minnett et al., 2011). Reported TBL thicknesses can reach 8 mm (Ginzburg et al., 1977), but generally refer to the upper 1 mm (Donlon et al., 2002).

Mesocosm experiments offer an optimal compromise between controlled laboratory conditions and natural field environments, allowing for the resolution of the dimensions of the SML (Galgani et al., 2014). They enable the mechanistic understanding of processes through the application of technology that is not applicable in open ocean settings. In this study, we induced a phytoplankton bloom in a mesocosm setting to assess its effects on the SML and ULW in a multidisciplinary study (Bibi et al., 2025a). To investigate the effect the bloom has on the DBL and TBL, we applied a novel method to obtain high-resolution

in situ measurements of the oxygen concentration and temperature across the SML. Using a microprofiler equipped with microsensors, we continuously obtained in situ microprofiles of oxygen concentration and temperature from the air across the DBL/TBL into the ULW. We analysed night-time data without wind to ensure a focused assessment of the phytoplankton bloom's influence, independent of turbulent conditions or diurnal warming effects. The analysis of microprofiles provides in situ measurements of oxygen and temperature gradients, oxygen exchange rates, and DBL and TBL thicknesses, while

assessing the impact of the bloom. To our knowledge, this approach provides the first real DBL and TBL thicknesses representative of calm water masses at night conditions.

## 2 Methods

### 2.1 The mesocosm study

The mesocosm study was conducted between 15 May 2023 and 16 June 2023 at the Sea sURface Facility (SURF), located at

the Institute for Chemistry and Biology of the Marine Environment (ICBM) in Wilhelmshaven, Germany (Bibi et al., 2025a). SURF is an 8.5 m x 2.0 m x 0.8 m large basin with a retractable roof, which can be filled with seawater from the Jade Bay, North Sea (Gassen et al., 2024). The multidisciplinary mesocosm study, part of the Biogeochemical processes and Air-sea exchange in the Sea-Surface microlayer (BASS) project, aimed to investigate the effects of a phytoplankton bloom on processes in the SML and is described in detail by Bibi et al. (2025a). The mesocosm was filled with filtered and skimmed

seawater to remove suspended particles. Flow pumps at the bottom of the basin slightly mixed the ULW, resulting in a horizontally uniform water body and preventing particles and organisms from settling on the bottom. Through the addition of different nutrients on 26 May, 30 May, and 01 June, a phytoplankton bloom was induced.

As part of this multidisciplinary approach, various properties of the SML and ULW were recorded. Water temperature and salinity were measured using two conductivity–temperature–depth (CTD) sensors (48M; Sea and Sun Technology, Germany)

at depths of approximately 2 cm and 40 cm below the surface. Surfactant concentrations in the SML and ULW, as concentrations of Triton equivalent (Teq), were measured in discrete samples using a Voltammetry technique (797 VA Computrace, including 863 Compact Autosampler, Metrohm, Switzerland) with a hanging drop mercury electrode (Ćosović and Vojvodić, 1987). The chlorophyll *a* concentration in the ULW (at 40 cm depth) was measured continuously using a FerryBox (-4H-Jena PocketBox, 4H Jena Engineering, Germany), which also measured oxygen concentration, temperature,



and salinity. Chlorophyll *a* measurements were corrected using discrete water samples. Bacterial abundances in the SML and ULW were measured in discrete samples. A comprehensive description of all these methods and the biogeochemical dynamics throughout the study can be found in Bibi et al. (2025a). Discrete samples of phytoplankton and protozooplankton were analysed from 250 mL Lugol-fixed samples by AquaEcology GmbH & Co. KG. For abundance determination, the samples were transferred to sedimentation chambers and allowed to settle overnight (Utermöhl, 1958; DIN EN 15972, 2011). Individual

cells were then counted under an inverted microscope (100x, 200x and 400x magnification). If possible, organisms were identified to species level or otherwise assigned to the genus or a higher taxonomic level.

## 2.2 The microprofiler setup

Continuous microprofiles across the SML were acquired using one oxygen (OX-200) and two temperature (TP-200)

microsensors mounted on a MicroProfiling System (UNISENSE, Denmark) (Fig. 1). The microsensors were mounted a few centimetres apart, with their tips facing upwards and close to the water surface, which was defined as a depth of 0 µm. To measure from the air, across the surface into the ULW, profiles were initiated at a height of 3,000 µm in the air and descended to a depth of 7,000 µm in the water, in increments of 125 µm, for optimal spatial and temporal resolution. At each step, the microsensors took three measurements, each lasting 10 seconds, and recorded the mean and standard deviation before

proceeding to the next step. The slow movement ensured reduced turbulence around the microsensors. Each profile took between 40 and 50 minutes, and profiling was performed continuously throughout the mesocosm study. To cover the time before, during and after the bloom, profiles from 22 May onward were further analysed. To exclude the effects of solar radiation and diurnal heating, only profiles from one hour after sunset to one hour before sunrise (approximately 22:30 to 04:00 local time) were analysed, yielding four to ten profiles per night and 141 in total. The roof of SURF was closed at night to exclude

wind and rain effects, allowing examination of only the phytoplankton bloom's influence on oxygen concentration and temperature, and their gradients across the SML.

The oxygen microsensor was calibrated before the study to ensure consistent data. Potential biases compared to a daily calibrated sensor were assessed separately, with no consistent offset being established (Appendix A). Furthermore, the microsensor measurements were compared to oxygen measurements of the FerryBox, and biases were minimal or explainable

by oxygen gradients across the ULW (Appendix B). Our work focuses on the surface oxygen gradients rather than absolute concentrations, prioritising data consistency for night-to-night comparability over absolute measurement accuracy. Based on the microprofiles, oxygen and temperature gradients across the SML were calculated, along with the thicknesses of the oxygen DBL and TBL (Sect. 2.3 and 2.4). To compare the oxygen and temperature gradients, as well as the DBL/TBL thicknesses, with chlorophyll *a* concentration and surfactant concentration, the mean of each parameter per night was calculated, checked

for normality, and examined for correlation using a Spearman correlation analysis. This analysis shows the correlation between non-normally distributed variables, with a correlation coefficient $r > 0.7$ and a p-value $< 0.05$ indicating strong correlations, $0.4 < r < 0.7$ and $0.05 < p < 0.10$ indicating moderate correlations and $r < 0.4$ and $p > 0.10$ indicating weak or no correlations.





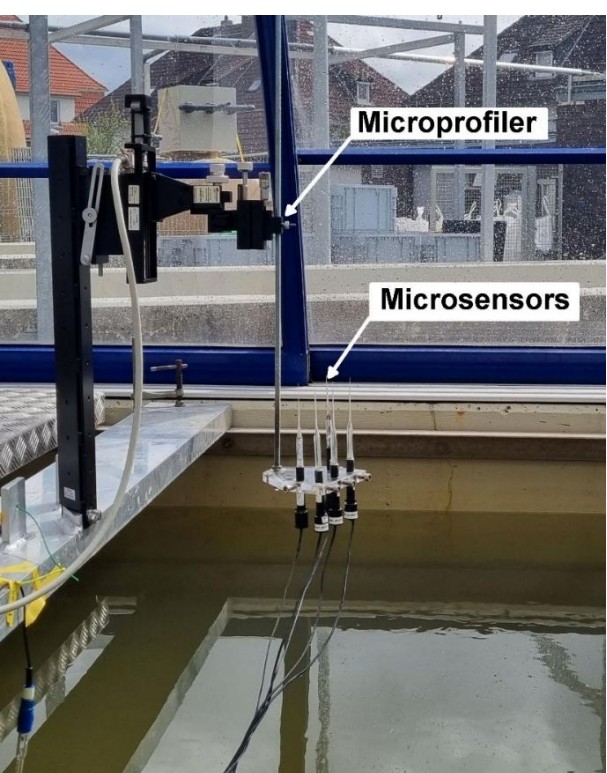

**Figure 1: Microprofiler setup. Microsensors mounted on a microprofiler above the SURF water basin.**

## 2.3 Determination of the diffusion boundary layer

To determine the DBL thickness and the oxygen gradients across the DBL, the mean of the triplicates at each depth was calculated (Fig. 2). The microprofiler's 0 µm depth (nominal air–water interface) did not necessarily reflect the sensor tip's true position, as manual alignment of multiple sensors introduced minor vertical offsets, which were corrected in our analysis. Six points (P1–6) in the profile were manually assigned. P1 and P2 were in the air above a sharp gradient, which indicated the location of the water surface. P2 was selected near the water surface, with a nearly linear trend between P1 and P2. P3 and P4 marked the upper and lower ends of the DBL, respectively. The DBL could be found by a large oxygen concentration gradient or by enhanced standard deviations and different gradients compared to the air and the ULW. P5 and P6 were assigned to the ULW below the sharp DBL gradient, where a linear downward trend was present.

Based on Rahlff et al. (2019), three linear regressions were computed between pairs P1 and P2, P3 and P4, and P5 and P6. The intersection between the air (P1 and P2) and DBL (P3 and P4) regression lines was the upper position of the water surface $H_0$. The intersection between the regression lines of DBL and ULW (P5 and P6) was the depth of the lower boundary of the DBL ($H_{DBL}$). The difference in depth between $H_{DBL}$ and $H_0$ was the DBL thickness, and the difference in oxygen concentration



between $H_{DBL}$ and $H_0$ was the oxygen gradient across the DBL. A positive gradient indicated higher oxygen concentrations in the ULW than in the air.

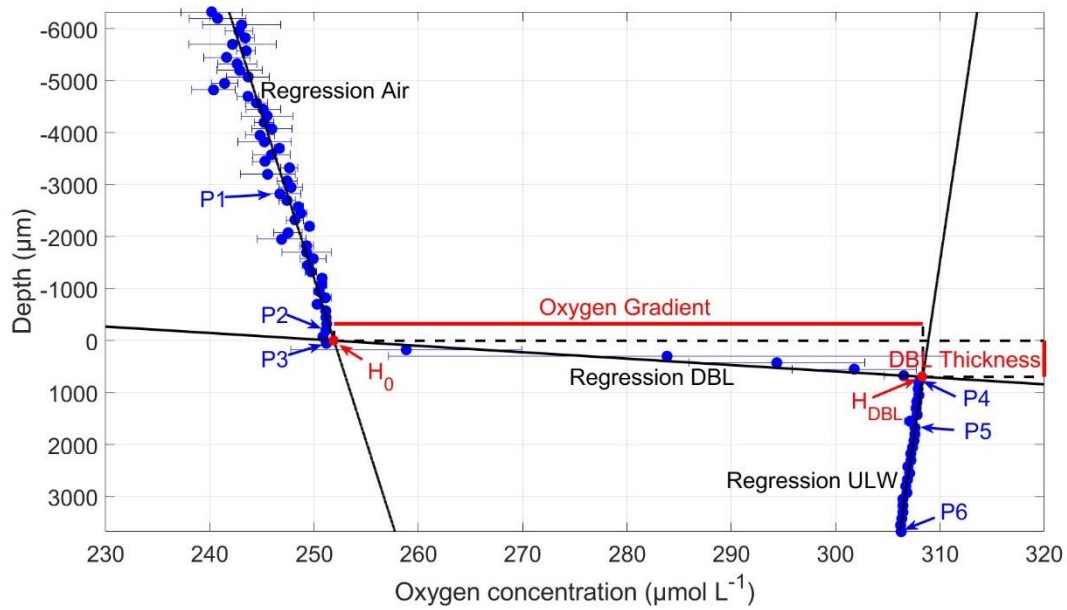

**Figure 2: Microprofile of the oxygen concentration (31 May 2023, Profile 2). Depth corrected for the proper sensor position, 0 µm**
**indicates air–water interface; points P1–P6 were used to calculate the regressions for air, DBL, and ULW. The intersection of air and DBL regression $H_0$ is the upper position of the water surface and the intersection of DBL and ULW regression $H_{DBL}$ is the lower DBL boundary. The vertical difference between $H_{DBL}$ and $H_0$ represents the DBL thickness and the horizontal difference represents the oxygen gradient.**

**2.4 Determination of the thermal boundary layer**

The analysis of the temperature microprofiles followed the same steps as the oxygen microprofile analysis (Fig. 3). The difference between the TBL and ULW was apparent in the profiles, as the TBL showed a pronounced temperature gradient. In contrast, the temperature in the ULW remained nearly constant with depth. To differentiate the air and TBL, which had similar temperature trends, the standard deviation at each step in the profile was considered. Temperatures across the TBL
were less variable than air temperatures, exhibiting apparent differences in standard deviations. This was used to identify the change from air to TBL and aided in assigning points P2 and P3. The TBL thickness was calculated as the difference between the depth of the lower TBL $H_{TBL}$ and the position of the water surface $H_0$. Contrary to the oxygen profiles and due to the nomenclature referring to a "cool skin layer" (Robertson and Watson, 1992; Soloviev and Schüssel, 1994; Wurl et al., 2018; Yan et al., 2024), a negative temperature gradient refers to a cooler upper boundary than lower boundary of the TBL.






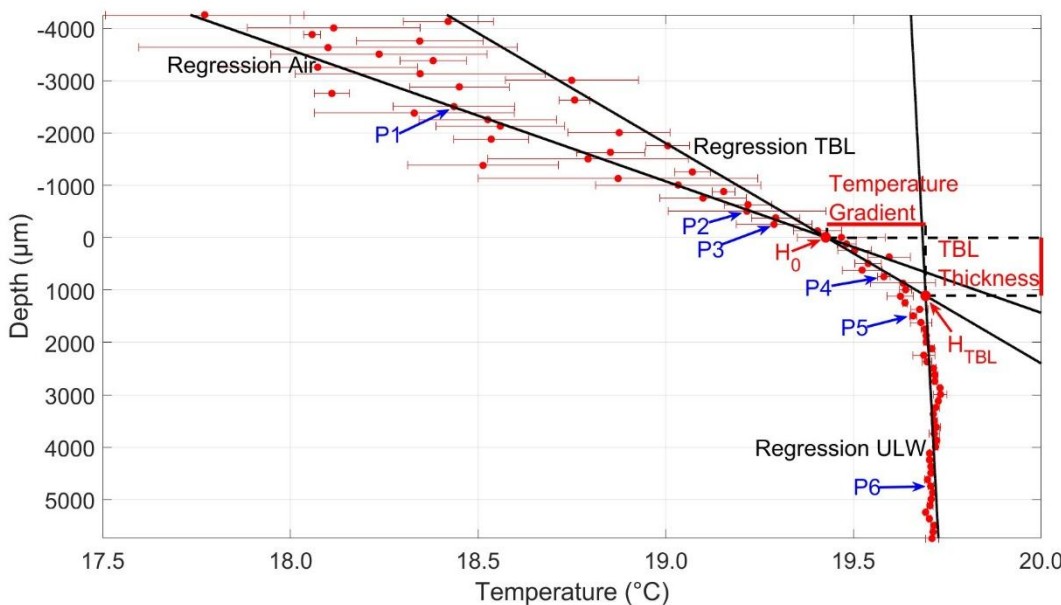

**Figure 3: Microprofile of the temperature (31 May 2023, Profile 1). Depth corrected for the proper sensor position, 0 µm indicates air–water interface; points P1–P6 were used to calculate the regressions for air, TBL, and ULW. The intersection of air and TBL regression $H_0$ is the upper position of the water surface and the intersection of TBL and ULW regression $H_{TBL}$ is the lower TBL boundary. The vertical difference between $H_{TBL}$ and $H_0$ represents the TBL thickness and the horizontal difference represents the temperature gradient.**

## 2.5 Determination of the gas exchange rate

The oxygen microprofiles were used to calculate the gas exchange rate, which represents the overall decline of oxygen at the lower end of the DBL. It was calculated after Eq. (1) by Goldman et al. (1988)

$$K = \frac{V}{A} * \frac{1}{\Delta t} * \ln(\frac{C_0}{C_t}) \tag{1}$$

where $K$ [cm h$^{-1}$] is the gas exchange rate, $V$ [cm$^3$] is the water volume and $A$ [cm$^2$] the surface area of SURF, $C_0$ [µmol L$^{-1}$] is the oxygen concentration at $H_{DBL}$ in the first profile of the night, $C_t$ [µmol L$^{-1}$] is the oxygen concentration at $H_{DBL}$ in the last profile of the night, and $\Delta t$ [h] is the time difference between the first and last profile. One gas exchange rate was calculated per night. Furthermore, the oxygen saturation of the water was calculated (Appendix C). To assess the phytoplankton bloom's impact on the gas flux between water and air, the diffusive rate was calculated (Appendix D). The diffusive rate indicates oxygen exchange between water and air for each profile, while the gas exchange rate represents the overall loss of oxygen in the water per night.



## 2.6 Horizontal temperature measurements with an infrared camera

Infrared (IR) imagery is capable of resolving horizontal temperature features across small eddies or slick edges (Marmorino et al., 2018). While studies of horizontal surface temperature gradients span local to global scales (Katsaros and Soloviev, 2004; Tozuka et al., 2018; Wurl et al., 2018; Zappa et al., 2019), analyses of the surface temperature on centimetre scales are mainly conducted in laboratory setups (Veron and Melville, 2001; Jessup et al., 2009; Wells et al., 2009). A FLIR SC7750-L infrared camera (Teledyne FLIR LLC, USA) was used to record horizontal small-scale temperature structures during this study. The

camera, mounted near the microsensors, which were positioned just outside the field of view, provided a resolution of $640 \times 512$ pixels, with each pixel covering an area of 1.657 mm $\times$ 1.586 mm, and a total field of view of 1.061 m $\times$ 0.812 m. Images were recorded at a rate of one image per second. While the absolute value had an accuracy of $\pm 1$ °C, the smallest temperature gradients could be detected, as the noise equivalent temperature difference was greater than 35 mK. Due to reflections, a reference image was created for correction by calculating the median from 60 images, which was subtracted from

each original image. A two-dimensional moving median filter was used to remove small outliers from each image. The filter determined the median for each pixel in the image by examining a $3 \times 3$ window of surrounding pixel values. This median then replaced the original pixel value, becoming the new output pixel.

## 3. Results

To assess the influence of the phytoplankton bloom on the oxygen diffusion boundary layer (DBL) and thermal boundary layer

(TBL), we first illustrate examples of oxygen microprofiles from different phases of the bloom. Then, we present the oxygen gradients and DBL thicknesses, comparing them to the concentrations of chlorophyll $a$ and surfactants. Next, we present the gas exchange rate and its influence by the phytoplankton and bacterial blooms. We subsequently present the temperature gradients and TBL thicknesses in comparison to the concentrations of chlorophyll $a$ and surfactants. Finally, we give an example of an infrared camera image of the surface temperature to assess horizontal temperature gradients.

## 3.1 Oxygen microprofiles

The mesocosm study was categorised into three distinct phases based on the dynamics of the chlorophyll $a$ concentration: the pre-bloom phase (18 May to 26 May), the bloom phase (27 May to 04 June), and the post-bloom phase (05 June to 16 June) (Bibi et al., 2025a). During the bloom and post-bloom phases, the water surface exhibited slick-like properties. For the oxygen microprofiles during the nights of all bloom phases, the oxygen concentration in both the air and the water decreased from one

profile to the next, with the larger decrease in the water (Fig. 4). During all nights, oxygen concentrations decreased with depth in the ULW, and a gradient in oxygen concentration existed in the air, with higher concentrations near the water surface. Typically, gradients in the air reached nearly 10 µmol L$^{-1}$ over a few millimetres. On 22 May, before the bloom, the oxygen gradient across the DBL was consistently negative. It intensified overnight with the oxygen concentration in the water decreasing by 10 µmol L$^{-1}$ (Fig. 4a). On 31 May, during the bloom, the oxygen gradient across the DBL remained constantly



positive (Fig. 4b). The oxygen concentration in the ULW was 48 µmol L⁻¹ higher than before the bloom. The decrease overnight of 28 µmol L⁻¹ was likewise greater. On 11 June, after the bloom, the oxygen concentration in the ULW was 36 µmol L⁻¹ lower than during the bloom, but 12 µmol L⁻¹ higher than before the bloom (Fig. 4c). The oxygen gradient across the DBL shifted from positive to negative during the night and declined by 21 µmol L⁻¹.




**Figure 4: Oxygen microprofiles throughout the night of 22 May in the pre-bloom phase (a), 31 May in the bloom phase (b), and 11 June in the post-bloom phase (c), depth corrected for the proper sensor position. 0 µm indicates the air–water interface. Times are the mean times of the profile; one profile took between 40 and 50 minutes to complete.**



## 3.2 Oxygen gradients

The oxygen gradients exhibited a trend over time similar to that of the chlorophyll *a* concentration (Fig. 5). Before the bloom on 22 May, the chlorophyll *a* concentration was 1.5 µg L$^{-1}$, and the oxygen gradient was negative, with a median of –6.81 µmol L$^{-1}$ and a low variability. After the induction of the bloom on 26 May, chlorophyll *a* increased and reached a maximum of 11.2 µg L$^{-1}$ on 02 June, while during the bloom, the oxygen gradients and their variability also increased. On 31 May, a maximum gradient of 49.48 µmol L$^{-1}$ was observed. As the chlorophyll *a* concentration declined in the post-bloom phase to background levels, oxygen gradients also decreased to a minimum of –5.90 µmol L$^{-1}$, similar to pre-bloom levels, and stayed around zero in the later post-bloom phase. The mean oxygen gradient over the entire study was +7.28 ± 15.98 µmol L$^{-1}$, but differences were evident when comparing the bloom phase to the pre-bloom and post-bloom phases. In the pre-bloom phase, the mean oxygen gradient of –2.16 ± 5.53 µmol L$^{-1}$ was similar to that of the post-bloom phase (+2.07 ± 4.82 µmol L$^{-1}$), but differed significantly from the mean gradient during the bloom (+24.90 ± 14.51 µmol L$^{-1}$). The oxygen gradients showed a strong correlation with the chlorophyll *a* concentration (r = 0.755, p < 0.001). The surfactant concentration, which started to increase at the bloom's peak and reached its maximum after the chlorophyll *a* and oxygen gradient maxima, was not significantly correlated with them (r = 0.215, p = 0.300).

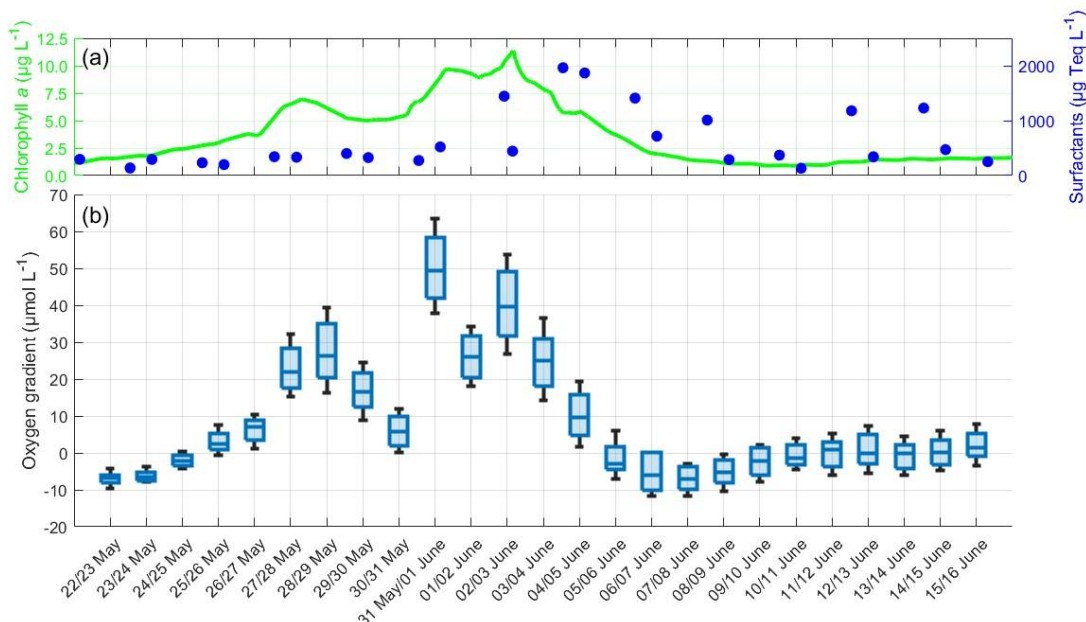

**Figure 5 (a): ULW chlorophyll *a* concentration (green) and SML surfactant concentration (blue) between 22 May and 16 June, (b): Oxygen gradients in the SML during the nights between 22 May and 16 June, 4–10 profiles per night, box: 25 % to 75 % quartile, horizontal line: median, whiskers: largest and smallest value.**



### 3.3 Oxygen diffusion boundary layer thickness

The thickness of the oxygen DBL showed a different trend than the oxygen gradients (Fig. 6). In the pre-bloom phase, the median DBL thickness varied within a narrow range (680 µm to 939 µm). Towards the end of the bloom phase, the DBL thickness increased, probably due to elevated surfactant concentrations. The surfactant concentration reached its maximum of 1963 µg Teq L$^{-1}$ on 04 June, and the DBL thickness reached its maximum, with a median of 1483 µm on 06 June. Subsequently, the surfactant concentration and DBL thickness decreased but remained higher than in the pre-bloom phase. Overall, DBL thicknesses showed no correlation with the chlorophyll $a$ concentration (r = –0.115, p = 0.584) or oxygen gradients (r = –0.272, p = 0.187), but a moderate correlation with surfactant concentrations (r = 0.490, p = 0.014) was observed. Although the mean DBL thickness (mean entire study: 936.73 ± 369.49 µm) increased from the pre-bloom phase (831.61 ± 344.32 µm) to the bloom phase (911.81 ± 342.82 µm) and post-bloom phase (1012.31 ± 393.76 µm), this increase was not significant.

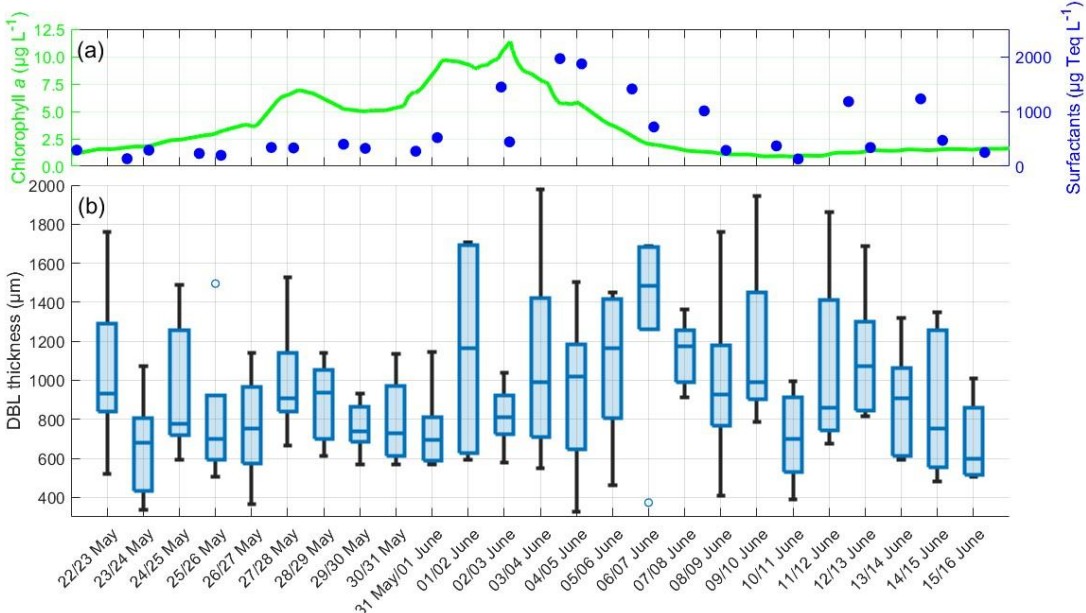

**Figure 6 (a): ULW chlorophyll *a* concentration (green) and SML surfactant concentration (blue) between 22 May and 16 June, (b): Oxygen DBL thickness during the nights between 22 May and 16 June, 4–10 profiles per night, box: 25 % to 75 % quartile, horizontal line: median, whiskers: largest and smallest nonoutlier value, open circles: outliers (difference to next value > 1.5 times interquartile range).**

### 3.4 Gas exchange rate

The night-time gas exchange rate followed the trends of chlorophyll *a* concentration and phytoplankton abundance during the pre-bloom and bloom phases (Fig. 7). In the post-bloom phase, it appeared to be influenced by bacterial abundance. The




phytoplankton abundance increased steadily until 03 June with a maximum of approximately 160 million cells L⁻¹, before it began to decline. The protozooplankton abundance began to increase after 31 May, reaching its maximum of 39 million cells L⁻¹ on 04 June. The bacterial abundances were 1.11 x 10⁹ cells L⁻¹ (SML) and 0.69 x 10⁹ cells L⁻¹ (ULW) on 22 May. There was a slight decrease in abundance during the bloom, but an increase in both the SML and ULW in the post-bloom phase, with their maxima on 14 June with 1.47 x 10⁹ cells L⁻¹ (SML) and 1.68 x 10⁹ cells L⁻¹ (ULW).

Before the bloom phase, the gas exchange rate reached its minimum at 0.44 cm h⁻¹ on 23 May, it then increased to a maximum of 1.91 cm h⁻¹ on 31 May during the bloom, before declining after the bloom to approximately 1.20 cm h⁻¹. The higher gas exchange rate in the post-bloom phase compared to the pre-bloom phase was accompanied by a higher bacterial abundance, both in the SML and ULW. In contrast, the phytoplankton abundances remained similar compared to pre-bloom levels. The mean gas exchange rate over the entire study was 1.21 ± 0.38 cm h⁻¹. In the pre-bloom phase, the mean gas exchange rate was

0.58 ± 0.11 cm h⁻¹, before increasing to 1.49 ± 0.32 cm h⁻¹ in the bloom phase, and then decreasing to 1.20 ± 0.18 cm h⁻¹ in the post-bloom phase. The gas exchange rate showed a moderate correlation with the phytoplankton abundance (r = 0.559, p = 0.004), but not with zooplankton abundance (r = 0.324, p = 0.114) or chlorophyll *a* concentration (r = 0.384, p = 0.059). The oxygen saturation followed a similar trend to the gas exchange rate, being influenced by both the phytoplankton and bacterial abundances (Appendix C). The diffusive rates exhibited a trend similar to that of the oxygen gradients, with a

significant increase during the bloom phase, but comparable levels in the pre-bloom and post-bloom phases (Appendix D).

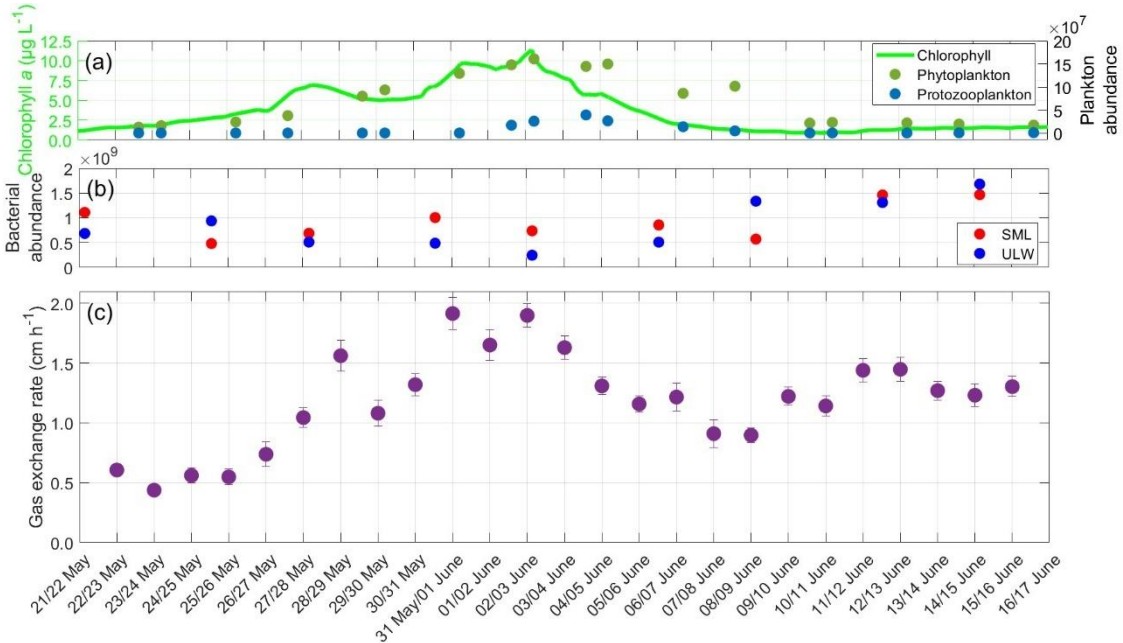

**Figure 7 (a): ULW chlorophyll *a* concentration (green), phytoplankton abundance (dark-green), and protozooplankton abundance (blue) between 22 May and 16 June, (b): Bacterial abundance in the SML (red) and ULW (blue), (c): Mean gas exchange rate during 290 the nights.**





## 3.5 Temperature gradients

The temperature microprofiles observed during the night appear similar across all nights (Appendix E). The temperature gradients did not follow a trend for either sensor, neither in their median nor in their variability (Fig. 8). A cooler skin layer

was always present, with negative temperature gradients approaching –0.4 °C in extreme cases. There were no correlations between the temperature gradients for both sensors and either the chlorophyll $a$ concentration (Sensor 1: r = 0.067, p = 0.439; Sensor 2: r = –0.077, p = 0.374) or the surfactant concentration (Sensor 1: r = 0.224, p = 0.003; Sensor 2: r = –0.084, p = 0.331). The differences in mean temperature gradients between both sensors ranged from 0.001 °C to 0.142 °C, with a mean of 0.048 ± 0.036 °C. The mean temperature gradient over the entire study, as measured by both combined sensors, was

–0.133 ± 0.079 °C. No significant differences were found between the pre-bloom (–0.130 ± 0.073 °C), bloom (–0.137 ± 0.084 °C), and post-bloom (–0.132 ± 0.079 °C) phases.

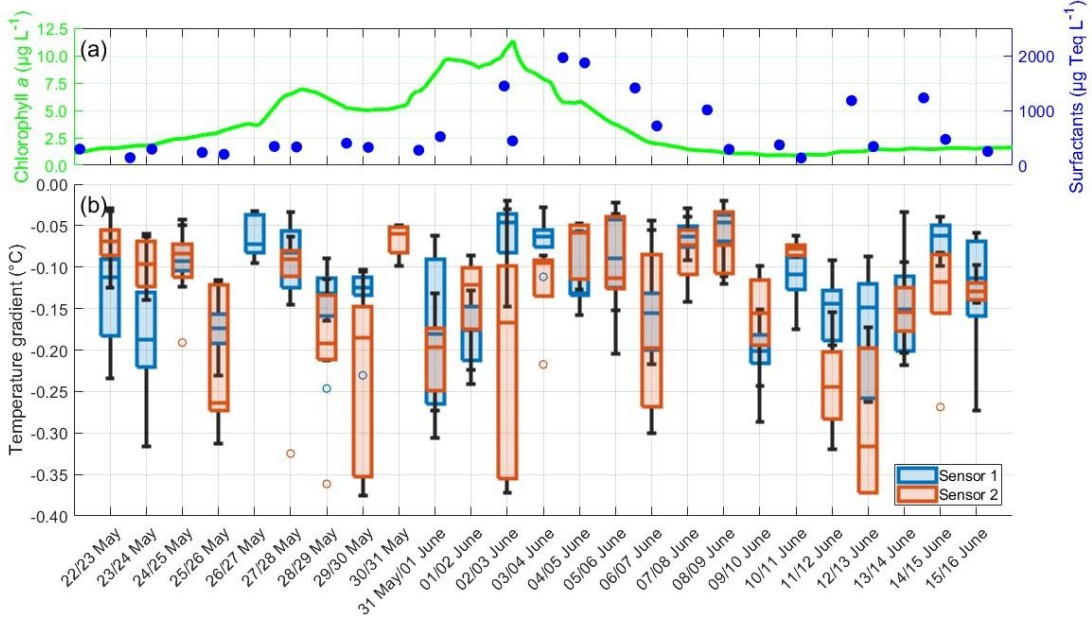

**Figure 8 (a): ULW chlorophyll *a* concentration (green) and SML surfactant concentration (blue) between 22 May and 16 June, (b):**
**Temperature gradients from Sensor 1 (blue) and Sensor 2 (red) during the nights between 22 May and 16 June, 4–10 profiles per night, box: 25 % to 75 % quartile, horizontal line: median, whiskers: largest and smallest nonoutlier value, open circles: outliers (difference to next value > 1.5 times interquartile range).**





### 3.6 Thermal boundary layer thickness

The TBL thickness showed no trend but exhibited fewer extreme values than the temperature gradients (Fig. 9). There was no correlation with the chlorophyll *a* concentration (Sensor 1: r = –0.100, p = 0.243, Sensor 2: r = –0.112, p = 0.195) or the surfactant concentration (Sensor 1: r = –0.082, p = 0.338; Sensor 2: r = 0.123, p = 0.153). However, the TBL thickness showed a moderate negative correlation with the temperature gradients (Sensor 1: r = –0.453, p = 0.037; Sensor 2: r = –0.615, p = 0.002). The differences in mean TBL thickness between the sensors ranged from 4.97 µm to 652.13 µm, with the mean

difference being 271.70 ± 190.71 µm. The mean TBL thickness over the entire study, as measured by both combined sensors, was 1299.50 ± 391.80 µm, being approximately 360 µm thicker than the oxygen DBL. There were no significant differences in the mean TBL thickness of the pre-bloom phase (1239.42 ± 342.76 µm), bloom phase (1277.09 ± 360.34 µm), and post-bloom phase (1348.41 ± 393.76 µm).

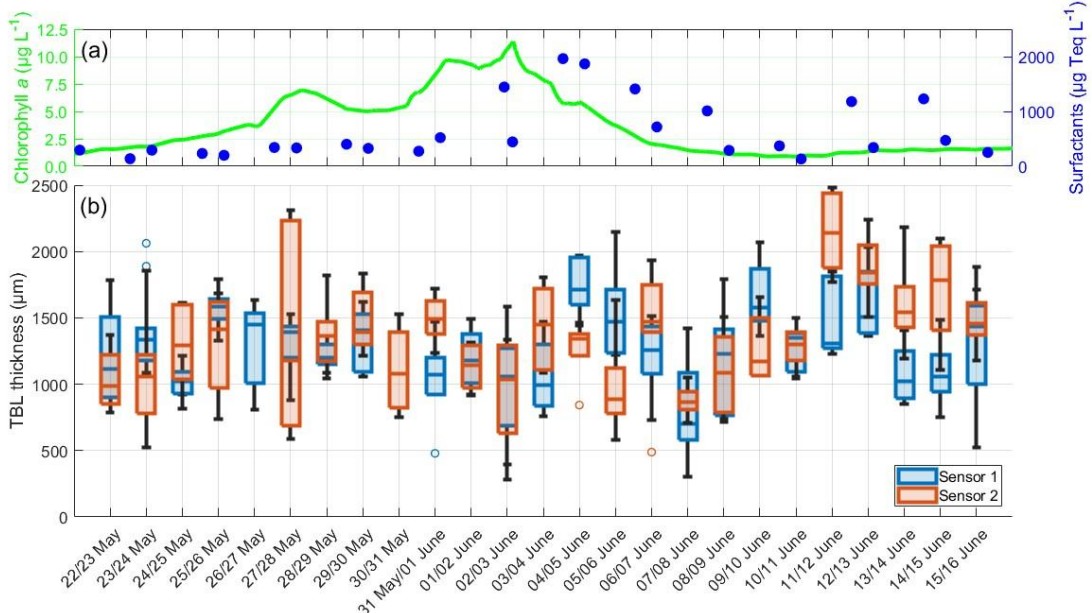


**Figure 9 (a): ULW chlorophyll *a* concentration (green) and SML surfactant concentration (blue) between 22 May and 16 June, (b): TBL thickness from Sensor 1 (blue) and Sensor 2 (red) during the nights between 22 May and 16 June, 4–10 profiles per night, box: 25 % to 75 % quartile, horizontal line: median, whiskers: largest and smallest nonoutlier value, open circles: outliers (difference to next value > 1.5 times interquartile range).**


### 3.7 Horizontal temperature gradients

The IR camera image from the post-bloom phase on 11 June revealed a heterogenous temperature distribution, highlighting distinct horizontal temperature gradients across the field of view, which measured 1.061 m × 0.812 m (Fig 10a). Zooming in,



the two white lines show examples of strong gradients (Fig. 10b). The upper-left line indicates a temperature difference of

0.232 °C over a distance of 32.2 mm. The lower-right line indicates a temperature difference of 0.224 °C over a distance of

22.1 mm. Both examples illustrate the potential for substantial horizontal temperature variations within a small observed area

of a few centimetres. These horizontal surface temperature differences support the observed temperature differences measured

by both temperature microsensors. On 11 June, the two microsensors, mounted a few centimetres apart, recorded a mean

difference in the temperature gradient of 0.091 ± 0.099 °C, which falls well within the range of the horizontal temperature

gradients detected by the IR camera.

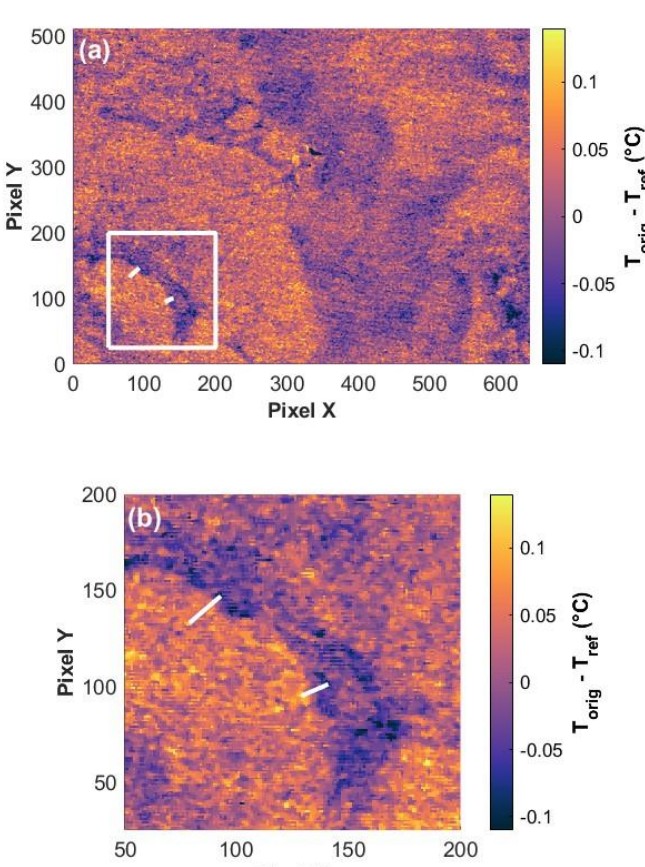

**Figure 10 (a): Corrected IR camera image ($T_{orig}$–$T_{ref}$) from 11 June at 21:00:25 (field of view = 1.061 m x 0.812 m, 100 pixels X = 0.166 m, 100 pixels Y = 0.159 m). The white square in (a) shows the zoomed view in (b): The two lines demonstrate two examples**

**where, despite a small distance of 32.181 mm (top left) and 22.087 mm (bottom right), high temperature differences of 0.232 °C (top left) and 0.224 °C (bottom right) were measured.**





## 4. Discussion

### 4.1 Measurements of the DBL and TBL

The use of microsensors and a microprofiling system provides a comprehensive dataset of in situ measurements of oxygen and temperature gradients across the SML over several weeks in a mesocosm study. The high-resolution microprofiles have enabled the direct assessment of the DBL and TBL thicknesses. To investigate the effect of the phytoplankton bloom on these properties solely, night-time microprofiles were analysed, and atmospheric influences such as solar radiation, diurnal warming and wind forcing on the water surface were excluded. Both oxygen and temperature microprofiles provide insights into the

processes and properties of the SML, such as the thicknesses of diffusion and thermal boundary layers, oxygen and temperature gradients across these layers, and gas exchange rates, which were previously measured indirectly.

   The study of air–sea gas exchanges has long been prevalent, from the proposal of a two-film model by Whitman (1923) to many other studies and concepts (Danckwerts, 1951; Liss and Slater, 1974; Deacon, 1977; Woolf, 1997; Asher et al., 2004; Zappa et al., 2007). Our in situ oxygen microprofiles, measured under non-turbulent conditions, reveal a three-layer structure

at the air–sea interface: two millimetre-thick layers in the air and ULW with notable oxygen gradients, and one sharp diffusion boundary layer right at the air–sea interface, which connects both. Observations of air flow by Buckley and Veron (2016) revealed layers of reduced turbulence directly above the water surface, which are prevalent under various atmospheric conditions. Our measurements indicate that significant gradients in gas concentration are present in these layers. Since gas flux measurements are often performed at heights of several metres above the water surface (Rutgersson et al., 2016; Wanninkhof

et al., 2019), the presence of these millimetre-scale gradients near the surface leads to uncertainties in the parameterisation of surface gradients and gas fluxes.

   Our measurements show that the DBL thickness (936.73 ± 369.49 µm) is significantly larger than the 50 µm thickness of diffusion sublayers defined by GESAMP (1995). It is, however, close to their definition of the viscous sublayer thickness, which is 1000 µm for a layer of reduced turbulence. Our measurements align well with the 1100 µm thick DBL, Rahlff et al.

(2019) measured using microsensors under laboratory conditions. However, they are significantly larger than the 350 µm – 500 µm reported by Adenaya et al. (2021), who applied a similar technique but did not include phytoplankton in their study. For other gases, whose fluxes are also predominantly dependent on properties of the water, like $CO_2$ (Liss, 1973; Ribas-Ribas et al., 2018), a DBL thickness of 50 µm is assumed (Robertson and Watson, 1992). Our measurements of the DBL indicate that its thickness under calm conditions is substantially underestimated and instead falls within the region of the viscous

sublayer. While our study shows a case without turbulence, its findings are important to consider when calculating gas fluxes under calm conditions and underline the need for gas flux measurements as close as possible to the water surface.

   Studies for determining the TBL thickness and temperature gradients often rely on indirect measurements and remote sensing of the surface temperature, using physical principles based on the temperature anomaly across the SML and the sum of radiative heat fluxes (Saunders, 1967). While early studies deliver a wide range of TBL thickness estimates (Ginzburg et al., 1977),





more recent studies often assume a TBL thickness of around 1 mm (Donlon et al., 2002; Jaeger et al., 2025). We are now able to confirm these assumptions with in situ data, with a mean TBL thickness of 1299.5 ± 391.8 µm, which is significantly thicker than the 300 µm thick thermal surface layer reported by GESAMP (1995). Our in situ measurements of a temperature gradient of –0.133 ± 0.079 °C also confirm common observations of a cool skin layer around –0.1 to –0.2 °C (Donlon et al., 1999; Murray et al., 2000; Donlon et al., 2002; Minnett et al., 2011; Jaeger et al., 2025). Most of these observations, however, were

made in field conditions, where wind enhances the cool skin effect. Our measurements indicate that, under calm night-time conditions, a cool skin layer is always present, reaching values of more than –0.3 °C due to net heat loss. Common parametrisations of the cool skin effect, solely based on wind speed (Donlon et al., 2002; Minnett et al., 2011), are well-defined for higher winds but overestimate the cool skin effect in the absence of wind compared to our measurements (–0.44 °C and –0.77 °C). Thus, other parameters influencing the net heat flux, such as buoyancy fluxes or air temperature and humidity

(Soloviev and Schüssel, 1994; Fairall et al., 2003), must be considered when calculating cool skin effects at very low wind speeds. The heterogeneous temperature distribution we detected with the IR camera under slick conditions, which also occurs under rain conditions (Wurl et al., 2019), highlights the need to base these calculations not only on measurements from a single point, but on the mean from a larger area.

**4.2 Influence of the phytoplankton bloom**

The phytoplankton bloom has significantly different effects on the oxygen gradients compared to the temperature. The increased daytime production of oxygen during the bloom resulted in approximately a 20 % increase in oxygen concentration in the ULW, which persisted into the night and was gradually depleted through respiratory processes. This led to increased oxygen gradients across the DBL, and a strong correlation was observed with the chlorophyll $a$, which acted as a proxy for the bloom. The temperature gradients, however, showed no correlation to the chlorophyll $a$, and thus no direct impact of the bloom.

We suggest that the temperature gradients were limited by buoyancy fluxes, in which a decrease in surface temperature led to an increased density and a replacement of the surface water with less dense ULW (Soloviev and Schüssel, 1994). These buoyancy fluxes could not be resolved by the microprofiles, as they occur on scales of a few minutes (Wurl et al., 2019). They also impact the exchange of oxygen and other gases, as surface water, which becomes gradually depleted in oxygen due to diffusive fluxes with the atmosphere, is regularly replaced with more oxygen-rich underlying water (MacIntyre et al., 2002).

The thickness of the diffusion boundary layer is not directly related to the chlorophyll $a$ and shows no correlation with the oxygen gradients. It is, however, moderately correlated to the surfactant concentration. While the peak of the surfactant concentration is a result of the phytoplankton bloom, the higher levels during the post-bloom phase are a result of the subsequent bacterial bloom, as both are producers of surfactants (Žutić et al., 1981; Kurata et al., 2016). Surfactants are known to directly retain molecular exchanges between the ocean and atmosphere, leading to a thicker DBL and decreasing gas

exchange (Liss, 1977; Cunliffe et al., 2013; Ribas-Ribas et al., 2018; Mustaffa et al., 2020). On the contrary, the TBL thickness showed no correlation with the surfactants but a moderate correlation with the temperature gradients. This leads to the



assumption that the direct effect of surfactants in reducing air–sea exchanges may be more effective for gas exchanges, such as oxygen exchange, but less effective for energy exchanges, like heat exchange. The SML with varying concentrations of surfactants significantly impedes oxygen transfer along concentration gradients (Goldman et al. 1988; Wurl et al., 2011; Rahlff et al., 2019). In the case of the TBL, however, heat equilibrates much faster, because the thermal diffusion coefficient in water is by two magnitudes greater than the oxygen diffusion coefficient (Bindhu et al., 1998; Ambari et al., 2022). Therefore, temperature gradients equilibrate faster than oxygen concentration gradients, which can mask correlations between temperature gradients and surfactant concentrations in the SML.

The high gas exchange rate in the bloom and post-bloom phases shows the combined influence of both the phytoplankton bloom and the subsequent bacterial bloom, as both lead to a decrease in oxygen concentration during the night through heterotrophic processes. Furthermore, we observed a constantly positive gas exchange rate and a reduction in oxygen concentration throughout the night, even when the diffusive rate was negative, indicating a diffusive increase in oxygen concentration. This underlines that biological processes in the mesocosm, not diffusive fluxes with the atmosphere, drove the oxygen concentration. This is supported by Rahlff et al. (2019), who also demonstrate that biological factors significantly outweigh the diffusive processes in oxygen concentration. They further demonstrate that biological processes in the ULW, rather than in the SML, drive the oxygen concentration at the water surface. In this study, chlorophyll *a* and phytoplankton abundance were only measured in the ULW but not the SML. The findings of Rahlff et al. (2019) provide confidence that no significant influence of biological processes within the SML was overlooked and that the processes measured in the ULW substantially control those in the SML.

## 4.3 Perspective: DBL and TBL under oceanic conditions

While the influence of wind was excluded in the analysis to focus on the impact of the phytoplankton bloom, it is essential to note that wind would have a significant effect on the DBL and TBL, leading to near-surface turbulent mixing (Liss, 1973; Ginzburg et al., 1977; Donlon et al., 1999; Murray et al., 2000; MacIntyre et al., 2002). These turbulent conditions may lead to enhanced exchanges between the SML, ULW, and atmosphere, resulting in smaller surface gradients and thinner, less distinct surface layers (Sromovsky et al., 1999; Wurl et al., 2011). Surfactants have both a direct effect on air–sea exchanges by retaining molecular exchanges at the surface and an indirect effect by reducing turbulence and damping waves (Liss, 1977; Goldman et al., 1988; Laxague et al., 2024). While this study shows the direct effects of surfactants retaining exchanges across the water surface, the presence of wind would enhance the indirect effects on gas and heat exchanges. Thus, the bloom's impact on surface properties might be disproportionally high under higher wind conditions, as it increases surfactant production and reduces the wind-driven turbulence.

Although the microsensor measurements were only taken at one point inside the mesocosm, the slow movement of the water ensured that a larger water mass was measured, making the measurements representative of the entire mesocosm. At each vertical step, measurements were integrated over a 30-second period, and a complete profile required 40–50 minutes to record.



Consequently, the profiles reflect mean conditions during this interval rather than instantaneous values, and fast surface
processes, such as buoyancy fluxes, could not be resolved. Although wind would make microsensor measurements more field-realistic, the resulting waves introduced high variability into the profiles. This variability is especially pronounced near the surface, making it challenging to accurately determine gradients and boundary layer thicknesses.

The strong influence of phytoplankton blooms may have global implications, as slicks, often formed by these blooms, cover large parts of the oceans and have significant impacts on air–sea gas exchange (Goldman et al., 1988; Romano, 1996; Mustaffa
et al., 2020). The effect of these blooms on the SML temperature, while not pronounced under no-wind conditions, is potentially relevant for global heat exchanges under typical oceanic wind conditions. Wurl et al. (2018) have already shown that natural slicks can have a significant impact on the temperature and salinity of the SML, indicating retention of evaporation. This mesocosm study further shows the need to improve cool skin parametrisations for low winds. Uncertainties in these parametrisations can have considerable implications, from validating satellite sea surface temperature measurements (Cronin
et al., 2019), to having an impact on gas fluxes with the atmosphere (Ward et al., 2004; Yang and Langdon, 2025) and the calculation of air–sea heat and freshwater fluxes, impacting the weather and climate (Grist et al., 2016; Zhao and Knudson, 2024).

While the use of microsensors under conditions with wind proved difficult, more in situ data of the DBL and TBL under varying wind conditions are needed, not only to assess the wind influence on the microprofiles, but also to investigate the
indirect effects of surfactants on gas and heat exchanges through their turbulence-reducing properties. The use of microsensors in our mesocosm study enabled us to gain a mechanistic understanding of how phytoplankton blooms affect the DBL and TBL. Our in situ measurements provide direct insights into the oxygen and temperature across the SML. These results represent a significant step in highlighting the importance of the SML for large-scale air–sea exchanges, based on in situ observations.

## 5. Conclusion

The mesocosm study, as part of an interdisciplinary study, provides high-resolution in situ measurements of oxygen concentration and temperature across the SML. Microsensors were used to obtain continuous in situ microprofiles from the air, through the SML, into the ULW, which were utilised to calculate surface gradients of oxygen and temperature, as well as the thicknesses of the oxygen diffusion boundary layer and thermal boundary layer. These in situ measurements revealed a strong correlation between the oxygen gradients and chlorophyll *a* concentration, while the DBL thickness was moderately
correlated with surfactant concentration. The blooms of phytoplankton and bacteria, through oxygen production, consumption, and surfactant production, heavily influenced the DBL. On the temperature gradients and TBL thickness, the phytoplankton bloom had no direct effect under wind-free conditions. With the inclusion of wind, the impact of surfactants on the DBL and TBL would likely increase, and the phytoplankton bloom would have an even greater impact on the SML properties. Overall, the microprofiles obtained by the microsensors proved valuable for gathering in situ data on previously unmeasurable
properties of the SML, while simultaneously assessing the effect of the phytoplankton bloom on these properties.





## Appendix A: Comparison of oxygen microsensors with different calibrations

In May 2024, a separate experiment was conducted inside SURF, where one daily-calibrated and temperature-corrected oxygen microsensor and one oxygen microsensor, which was only calibrated at the start of the experiment and not temperature-corrected, were compared. Both were mounted close to the inlet of a FerryBox (-4H-Jena PocketBox, 4H Jena Engineering, Germany) at approximately 30 cm water depth. The FerryBox measured the oxygen concentration (corrected with discrete samples analysed with the Winkler method) and water temperature. The results showed that daily calibration sometimes led to large jumps in oxygen concentration, depending on the temperature at the time of calibration (Fig. A1). No consistent offset was established between the daily calibrated sensor and the once-calibrated sensor. The offset of both microsensors compared to the FerryBox was similar. This experiment demonstrated that the lack of daily calibration likely did not lead to large offsets in the oxygen microsensor measurements obtained during the mesocosm experiment.

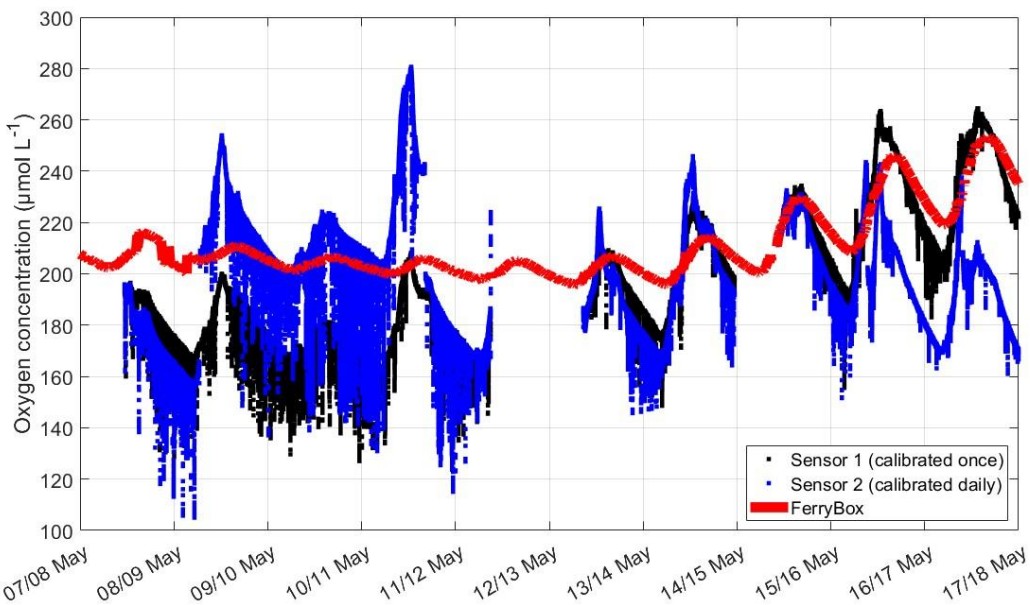

**Figure A1: Comparison of the oxygen concentration measured by an oxygen microsensor calibrated at the start of the experiment and not corrected for temperature (black), one microsensor calibrated daily and corrected for temperature (blue), and corrected oxygen concentration measured by a FerryBox (red), experiment conducted in May 2024 in SURF.**



## Appendix B: Comparison of the oxygen microsensor with the FerryBox

The oxygen concentration measured by the microsensor during the main mesocosm study was compared with oxygen measurements from a FerryBox (-4H-Jena PocketBox, 4H Jena Engineering, Germany), which were corrected using discrete
samples analysed with the Winkler method. The inlet of the FerryBox was located at approximately 40 cm depth. The comparison of the mean microsensor measurements from the ULW and FerryBox shows that both measurements were typically consistent, but with a time lag that could reach more than three hours (Fig. B1). Only in the post-bloom phase was a larger offset observed, as indicated by higher oxygen concentrations measured by the microsensors. These offsets can be attributed to a growing oxygen gradient in the ULW between the microsensors and the FerryBox, which can also be seen in the
microsensor profiles during this time (Fig. 4c). The results provide more confidence in the quality of the oxygen microsensor data, even if the microsensor was only calibrated once and not corrected for temperature.

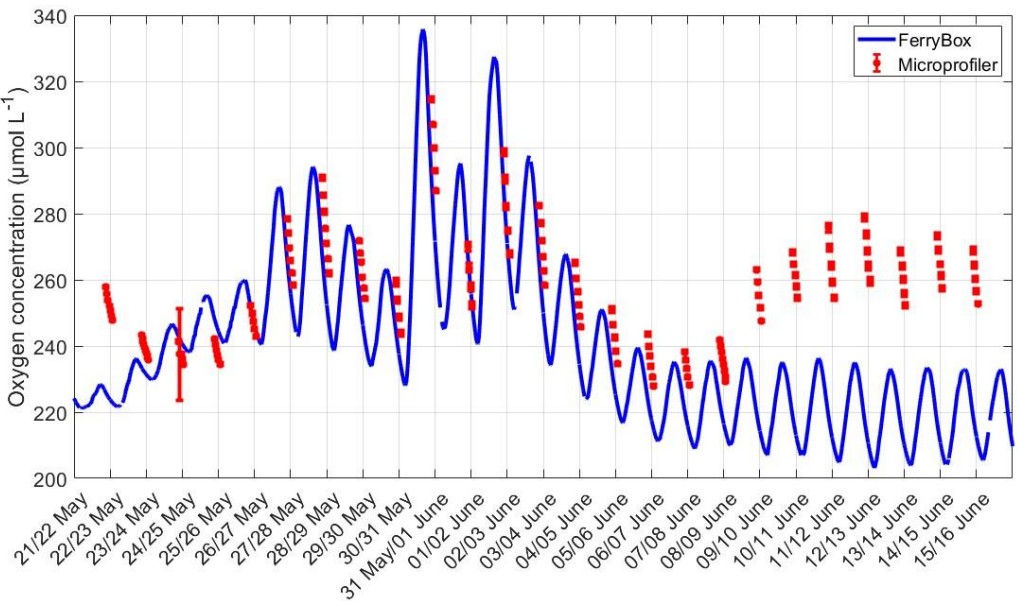

**Figure B1: Corrected oxygen concentration measured by the FerryBox at 40 cm depth (blue) and by the oxygen microsensor (red),**
**mean from the lower end of the DBL to the end of the profile.**

## Appendix C: Oxygen saturation

The microsensor profiles were used to compute the oxygen saturation of the surface water. The oxygen concentration and temperature measured by the microsensors below $H_0$, along with the salinity of the CTD at a depth of 40 cm, were used to



calculate the oxygen saturation profiles of the water using the Gibbs-Sea Water (GSW) Oceanographic Toolbox for MATLAB (McDougall and Barker, 2011). Differences compared to the calculation using salinity data from the CTD near the surface, which had a poorer data quality due to many outliers, were approximately 0.14% and not statistically significant.

The oxygen saturation trend was very similar to the trend in gas exchange rates (Fig. C1). It had a minimum of 96 % in the pre-bloom phase, then increased during the bloom phase to 127 %, and decreased in the early post-bloom phase to 101 %. Contrary to the trend in oxygen gradients, but similar to the trend in gas exchange rates, oxygen saturation increased again in the post-bloom phase to approximately 120 %. This also coincided with the post-bloom increase in bacterial abundance. The mean oxygen saturation over the entire study was $109.37 \pm 9.00$ %. Pre-bloom, it was $99.80 \pm 3.88$ %, rose during the bloom phase to $111.89 \pm 6.64$ %, and subsequently increased slightly to $112.35 \pm 8.94$ % in the post-bloom phase. Rising water temperatures during the mesocosm study (approximately a 7 °C increase from 18 May to 16 June; Bibi et al., 2025a) only partly explain this increase, as oxygen saturations were still 10 % higher post-bloom compared to pre-bloom when calculated with a constant temperature.

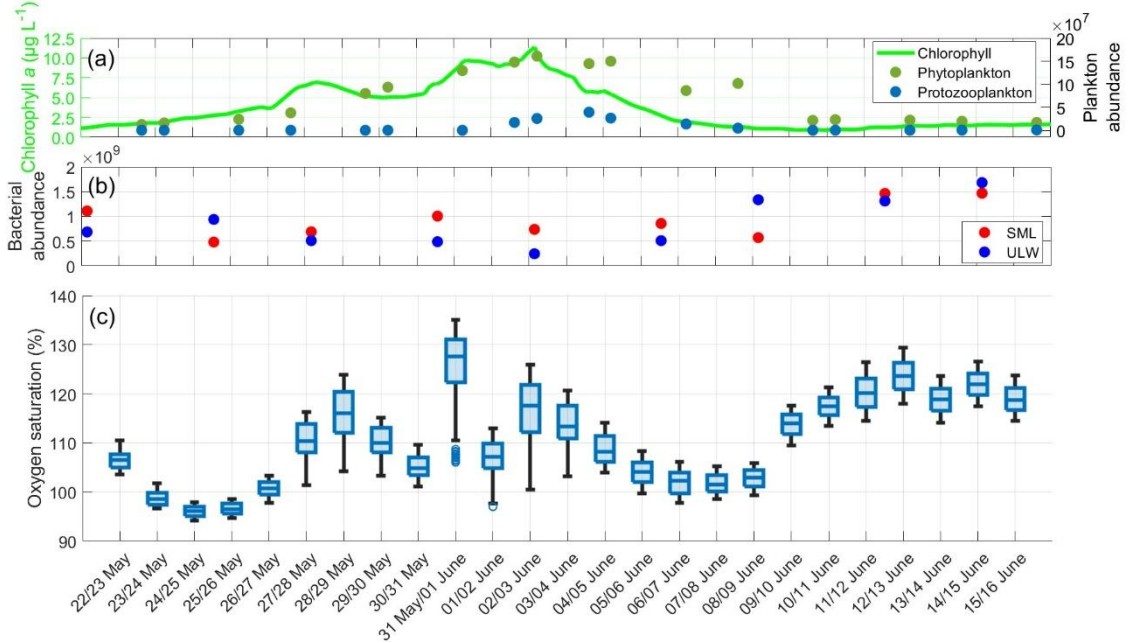

**Figure C1 (a): ULW chlorophyll *a* concentration (green), phytoplankton abundance (dark-green) and protozooplankton abundance (blue) between 22 May and 16 June, (b): Bacterial abundance in the SML (red) and ULW (blue), (c): Oxygen saturation during the nights, boxplots show all the oxygen saturation data in the DBL and ULW, box: 25 % to 75 % quartile, horizontal line: median, whiskers: largest and smallest value, open circles: outliers (difference to next value > 1.5 times interquartile range).**





**Appendix D: Oxygen diffusive rates**

525 In addition to the single gas exchange rate per night, the vertical diffusive rate for each profile during the night was calculated using Eq. (2) as described by Rahlff et al. (2019). It illustrates the vertical oxygen loss from water to air through diffusive processes in the absence of wind, while the gas exchange rate refers to the total rate of change in oxygen concentration during each night, regardless of driving factors.

$$DR = D * \frac{A}{V} * \frac{1}{\delta} * (C_{HDBL} - C_{H0}), \tag{2}$$

530 $DR$ [µmol L$^{-1}$ h$^{-1}$] is the vertical diffusive rate, and $D$ [cm h$^{-1}$] is the diffusion coefficient dependent on the temperature and salinity, which was obtained from the table of Ramsing and Gundersen (2000) using the microsensor temperature at $H_{TBL}$ and the salinity measured by the CTD in the ULW. $V$ [cm$^3$] is the water volume and $A$ [cm$^2$] the water surface area of SURF, $\delta$ [cm] is the DBL thickness, and $C_{HDBL}$–$C_{H0}$ [µmol L$^{-1}$] is the oxygen gradient in the DBL. Compared with Rahlff et al. (2019), the direction of the diffusive rate was adjusted to align with the definition of the oxygen gradient direction in this study. If the 535 diffusive rate was positive, oxygen diffused from the water into the air, and vice versa.

The diffusive rates of oxygen were strongly correlated with the chlorophyll $a$ concentration (r = 0.745, p < 0.001), but not with the surfactant concentration (r = 0.239, p = 0.250) (Fig. D1). The mean diffusive rate was +0.098 ± 0.225 µmol L$^{-1}$ h$^{-1}$, and there was a large increase during the bloom phase (+0.323 ± 0.243 µmol L$^{-1}$ h$^{-1}$) compared to the pre-bloom phase (–0.037 ± 0.081 µmol L$^{-1}$ h$^{-1}$) or post-bloom phase (–0.016 ± 0.060 µmol L$^{-1}$ h$^{-1}$). The trend in diffusive rates is very similar 540 to the trend in oxygen gradients, since the oxygen gradients are a major factor in calculating the diffusive rates. The diffusive rates show that in the bloom phase, oxygen mostly diffused from the water into the air (positive rate), while it mostly diffused from the air into the water in the pre-bloom and post-bloom phases (negative rate).





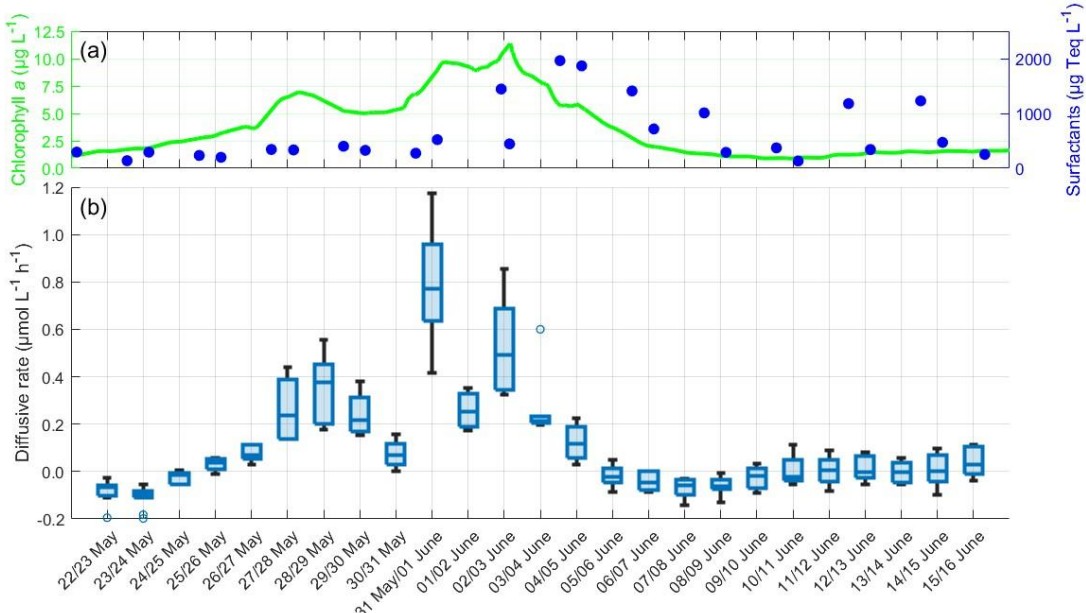

**Figure D1 (a): ULW chlorophyll *a* concentration (green) and SML surfactant concentration (blue) between 22 May and 16 June, (b): Oxygen diffusive rate in the SML during the nights between 22 May and 16 June, 4–10 profiles per night, box: 25 % to 75 % quartile, horizontal line: median, whiskers: largest and smallest value, open circles: outliers (difference to next value > 1.5 times interquartile range).**

## Appendix E: Temperature microprofiles during one night

The temperature microprofiles on 22 May give a typical example of the profiles during all nights (Fig. E1). The warmest profile was the first of the night, and all the subsequent profiles were approximately 0.05 °C cooler than the preceding profile. The air temperature decreased as it moved further away from the water surface; it also decreased from profile to profile during the night, and it was always lower than the water temperature. A cooler thermal boundary layer was present in all profiles. The only significant difference between the nights was the absolute temperature, which typically rose from night to night, starting at approximately 18.9 °C on 22 May and reaching around 22.6 °C on 15 June.





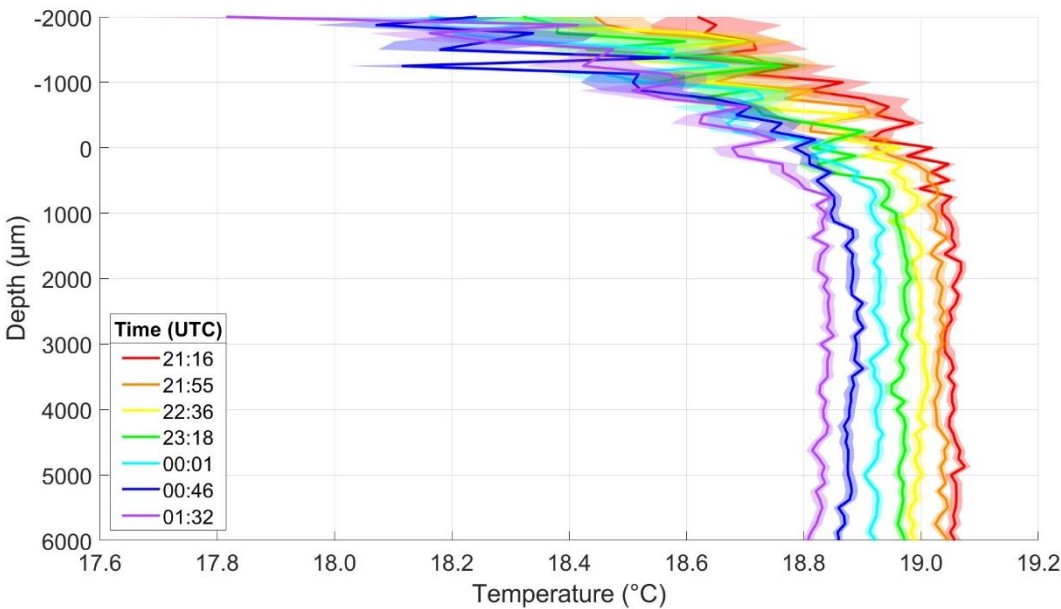

**Figure E1: Temperature microprofiles of Sensor 1 throughout the night of 22 May, in the pre-bloom phase, depth corrected for the**
**proper sensor position. 0 µm indicates the air–water interface. Times are the mean times of the profile; one profile takes**
**approximately 40 minutes to complete.**

### Data availability

The microsensor data from the entire mesocosm study are accessible for review at PANGAEA . They will be made publicly available by the end of 2025, as Rauch et al. (2025). Data from discrete samples during the  mesocosm study, like chlorophyll *a*, surfactants, and overall bacterial abundance, are available at PANGEA in Bibi et al. (2025b).

### Author contributions

CR analysed and interpreted all microsensor data and prepared the manuscript. LD contributed to the conduction of the
mesocosm study, including the implementation of the IR camera, analysis of the IR camera data, and contributed to the manuscript writing and revision. LJ and EFC-E contributed to the conduction of the mesocosm study, including the setup of the microsensors and microprofiler, and revised the manuscript. MR-R contributed to the organisation and conduction of the mesocosm study and majorly reviewed and revised the manuscript. JK contributed to the conduction of the mesocosm study, including the sampling and analysis of phytoplankton and zooplankton and revised the manuscript. AE revised the final





manuscript. OW co-designed the experimental setup, discussed with CR and MR-R the data analysis and interpretation, and majorly reviewed and revised the manuscript.

**Competing interests**

The authors declare that they have no conflict of interest.

**Acknowledgements**

We thank the German Research Foundation (DFG) for mainly funding the BASS project, as well as the Austrian Science Fund (FWF) for contributing to the funding of BASS SP1.2. We extend our gratitude to Riaz Bibi for coordinating the mesocosm study and the BASS project, for providing a comprehensive overview of the mesocosm study and its biogeochemical results,

and for contributing a significant portion of the discrete sample data. We are grateful to Isha Athale, Dmytro Spriahailo, Thorsten Brinkhoff, Thomas Reinthaler, and the BASS SP1.2 team for providing the bacterial abundance data and offering valuable insights into bacterial processes during the mesocosm study. We thank Michael Novak and Rüdiger Röttgers from BASS SP1.3, for providing the chlorophyll $a$ concentration data and thank Claudia Thölen and Jochen Wollschläger from BASS SP1.3, for providing the FerryBox data. We are grateful to Thomas Badewien and Janina Rahlff for providing valuable

input during the preparation of the manuscript. Finally, we thank everyone who conducted the mesocosm study, especially Carola Lehners and Michaela Gerriets, for their enormous effort in making this study possible.

**Financial support**

This research was supported by the project "Biogeochemical processes and Air–sea exchange in the Sea–Surface microlayer (BASS)", which was funded by the German Research Foundation (DFG) under Grant No 451574234. BASS subproject SP

1.2 was co-funded by the Austrian Science Fund FWF under the subproject number I 5942-N.



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
