# Peer review of "Phytoplankton blooms affect microscale gradients of oxygen and temperature across the sea surface microlayer"

_EGUsphere, 2025_

## Referee Comment (RC2)

Review of Rauch et al. Phytoplankton blooms affect microscale gradients of oxygen and temperature across the sea surface microlayer

The manuscript describes an experiment investigating the effects of a phytoplankton bloom on the DBL and TBL using an outdoor tank with a cover referred to as a mesocosm.  The measurements appear to be of high quality and the methods are sound.  The results showing strong correlation of the Oxygen parameters ("gradient,", DBL thickness, and k) and no correlation of the temperature parameters ("gradient" and TBL thickness) with Chlorophyl a and surfactant concentration are significant and worthy of publication.  The text is well written and organized.

Review criteria:

- Scientific significance: 3
- Scientific quality: 3
- Presentation quality: 3

The manuscript is suitable for publication after the authors address the following points to the satisfaction of the editor.

First of all, I am surprised that the authors would describe the difference between the measured parameters at the top and the bottom of the relevant BL as the gradient.  Based on the units, they are showing the difference, not the gradient.  This needs to be addressed and fixed.

My main concern is with the comparison of the TBL findings to corresponding published values for the bulk-skin temperature difference (dT) and TBL thickness based on field experiments.  The results are presented as calm night-time conditions but the apparent characteristics of the facility and the IR imagery suggests otherwise.

- The mesocosm appears to be an outdoor wave tank with a retractable roof which was used to exclude wind and rain effects.  The photograph in Fig. 1 is an inadequate representation of the setup and the reader should not have to go to other publications to figure out the facility details.  From the photos in the referenced publications of Bibi et al. and Gassen et al., it appears that the "roof" does not block off the openings at the ends of the tank.  If that is the case, then there may still be some effect of air movement over the tank. I suspect the ends are open since a dT was observed – if it was sealed then the water and air would eventually come into equilibrium and there would be no dT.
- The mesocosm included the use of an array of pumps at the bottom of the tank that continuously circulated water.  From Bibi et al. it appears that the circulation was horizontal, but I suspect that this subsurface forcing had some impact on the surface.  IR imagery of a calm surface with a net upward heat flux should clearly show convection cells – see publications by Saylor (web examples at https://cecas.clemson.edu/~jsaylor/saylor.rsch.irHiRes.html), which also address the issue of slicks.  The IR image in Fig 10 looks more like measurements in a wind flume at low

wind speed such as in publications by Jahne and colleagues, Veron, Marmorino and Smith, etc. The authors suggest that the appearance is due to slicks but they do not look like published images of slicks.

- There seems to be a significant difference between the measurements of the two temperature sensors. Sensor 2 shows a much larger quartile range in Fig. 8 that seems to vary with time (smaller pre- and post-bloom, larger during bloom.) The maximum difference of 0.142 C seems to be significant and should be addressed. The authors indicate that the horizontal differences in the IR imagery can explain this but the IR image is a snapshot of spatial features that presumably are varying in time while the temperature probes are average over time at a point. If the horizontal features change then it seems like they would be averaged out in time for the T sensors. I also don't think you can assume that the surface IR features are representative of the temperature in the TBL since the IR optical depth is O(10 microns).

- I think the comparisons of dT and the TBL thickness in sec. 4.1 to field results need to be less definitive. The authors state that their measurements confirm the TBL thickness of O(1 mm) reported by Donlon et al. and Jaeger et al., but both of those papers address field conditions. The authors also state that their measurements confirm the commonly cited -0.1 to -0.2 C dT but that result is for wind speeds above 6 m/s or so. I think it is not justified to state that the results show that the cool skin effect in the absence of wind are overestimated by the field measurements. I looked at Donlon et al and Minnett et al and could not find the implied values of -0.44 C and -0.77 C under calm conditions. If they came from scatter in the dT vs U plots in those refs that is not an appropriate comparison. The authors are correct that other factors besides wind affect dT, esp. heat flux. But unfortunately there is no information about the heat flux for the mesocosm. Was relative humidity measured? If so, you could compute the specific humidity difference and look at the dT vs dq correlation as in Yan et al. [2024 DOI: 10.1175/JPO-D-23-0103.1] since the latent heat flux likely dominated the dT forcing. The dT treatment would have been significantly improved by having heat flux measurements. I suggest using language like "consistent with" or "differs from" and avoid stating that this one simple, limited experiment under unquantified heat flux conditions confirms or shows errors in published field studies.

Other points

- Ward and Donelan (2006) measured the TBL thickness in the laboratory in a similar fashion and should be cited and compared with the results
- Zero lines on Figs 2-5 would be helpful
- How were the pre-, bloom, and post- periods determined? Vertical lines on Figs 5-9 showing these periods would be helpful
- The first sentence of Sec 3.5 states that all observed night profiles were similar and refers to Appendix E, but that appendix only shows profiles for one night.